# Digital Media Exposure and Health Beliefs Influencing Influenza Vaccination Intentions: An Empirical Research in China

**DOI:** 10.3390/vaccines10111913

**Published:** 2022-11-12

**Authors:** Qingting Zhao, Hao Yin, Difan Guo

**Affiliations:** 1School of Journalism and Communication, Shandong University, Jinan 250100, China; 2School of Journalism and Communication, Nanjing Normal University, Nanjing 210097, China; 3School of Journalism and Communication, Beijing Normal University, Beijing 100091, China

**Keywords:** digital media exposure, influenza vaccine, vaccination intentions, China

## Abstract

The purpose of this study was to investigate whether/how digital media exposure influences people’s intention to influenza vaccination. Through an anonymous online survey, we collected data on Chinese people’s exposure to influenza and influenza vaccine information on digital media platforms and their attitudes toward influenza vaccines (N = 600). The structural equation model analysis results strongly support to the research hypotheses and the proposed model. The findings reveal three major themes: (1) digital media exposure significantly influence the susceptibility and severity of influenza. (2) After exposure to digital media, it is helpful to understand the vaccine’s benefits, reduce the barriers to vaccination, and finally improve the intention to vaccination. (3) Users receive cues to action from digital media, and their vaccination intention tends to be positive. These findings explore how digital media exposure influences influenza vaccination intention and may provide insights into vaccine promotion efforts in countries. Research has shown that digital media exposure contributes to getting vaccinated against influenza.

## 1. Introduction

Influenza is a common respiratory disease that can spread from person to person and affects people of all ages, creating a significant global disease burden [1]. Prevention and control measures, especially influenza vaccination can effectively prevent the impact of influenza on people’s health [2]. In China, influenza vaccination rates are low except in some cities where the local government funds influenza vaccination programs because it is not included in the national immunization program and because people need to pay for their influenza vaccination [3,4]. However, China has a large population base. Considering the importance of the influenza vaccine to prevent influenza in humans, how to increase the intention of influenza vaccination has become a vital issue affecting people’s health [5].

Exposure to information about vaccines affects people’s willingness to vaccinate [6,7,8], as has been demonstrated in research on the HPV vaccine and COVID-19 vaccine willingness to vaccinate [9,10]. Although information exposure on social media can increase public knowledge about HPV and COVID-19 vaccines [11], exposure to negative information can also reduce people’s risk perceptions and lead to vaccine hesitancy, thus hindering vaccination [7,12,13,14]. The flu vaccine is no exception, and existing studies have come to different conclusions about how exposure affects people’s willingness to get an influenza vaccination. Some studies have found that people with more knowledge about influenza and its vaccination are more likely to get the flu vaccine [15,16,17]. In contrast, others have come to the opposite conclusion, suggesting that people’s exposure to social media messages about immunization side effects, for example, increases vaccine hesitation [18].

With the popularity of the Internet in China, the way people obtain information has changed dramatically, and digital media such as smartphones have become an important channel for people to get health-related information [10,19]. Digital media are gaining popularity not only among young people but also among the elderly [20]. Because of the critical influence of media messages on influenza vaccination, it is necessary to understand the impact of digital media exposure on people’s willingness to receive influenza vaccination. China’s most commonly used digital media platforms include Sina Weibo, Douban, QQ, Tieba, Zhihu, etc. [21].

The three-stage model of health promotion proposed by Street theoretically explains how media information affects people’s health behaviors. Specifically, the information that people are exposed to through various information sources shapes their attitudes towards health issues, influencing their health behaviors and outcomes [22]. However, people’s health decisions are influenced by psychological factors such as their beliefs, attitudes, and intentions [23,24]. The Health Belief Model (HBM) contains f perceived susceptibility, perceived severity, perceived benefits, perceived barriers, and cues to action. It can be used to study people’s beliefs, attitudes, and other psychological factors in the decision-making process of influenza vaccination [25]. Furthermore, related studies have shown that perceived susceptibility, perceived severity, perceived benefits, and cues to action in HBM positively influence influenza vaccination intentions, while perceived barriers negatively influence the intentions to the vaccination [1,6,26]. Therefore, based on the three-stage model, this study explores the influence process of “social media information exposure--attitudes toward influenza vaccination--the intentions to vaccinate.” In the stage of influenza vaccination attitudes, the variables reflecting health beliefs in the HBM model are introduced as mediators. Thus, a theoretical model is constructed to investigate the influence of the Chinese public’s social media exposure on influenza vaccination intentions.

Based on the theoretical model, this study proposes the following five hypotheses:

**H1.** 
*Digital media exposure positively affects perceived susceptibility and further positively influences vaccination intention mediated by perceived susceptibility.*


**H2.** 
*Digital media exposure positively affects perceived severity and further positively influences vaccination intention mediated by perceived severity.*


**H3.** 
*Digital media exposure positively affects perceived benefits and further positively influences vaccination intentions mediated by perceived benefits.*


**H4.** 
*Digital media exposure negatively affects perceived barriers and negatively influences vaccination intention mediated by perceived barriers.*


**H5.** 
*Digital media exposure positively affects cues to action and positively influences vaccination intentions mediated by action cues.*


## 2. Materials and Methods

### 2.1. Research and Design

This study investigates whether the Chinese public’s exposure to digital media information can influence the intentions to vaccinate against influenza through the mediating variable health beliefs. Therefore, this study builds a structural equation model to verify the validity of the above hypotheses and further explore the influence mechanism of information exposure to digital media on the intentions of the Chinese people to vaccinate against influenza.

### 2.2. Data Collection

This study obtained research samples through the network questionnaire survey. The questionnaire included basic demographic questions, the respondents’ digital media exposure, their perception of influenza vaccine (benefits or barriers, etc.) through digital media exposure, and their intentions to influenza vaccination. In this paper, more than 800 questionnaires were initially collected, and invalid questionnaires were eliminated through reliability tests. Subsequently, based on the age data from the 2010 official Chinese census [27], the questionnaires were quantified according to the age of the respondents so that the age distribution of the questionnaires matched the actual age distribution of Chinese people, and 600 valid questionnaires were finally obtained. The gender and age distribution of the valid questionnaire sample was relatively even.

More than half of the respondents had received a college education or above (37,562.5%), 86 had only received high school education (14.33%), 45 had only received primary/junior high education (7.50%), and 94 had received the highest education from vocational schools (15.67%). The questionnaire’s content did not involve medical or ethical issues, the respondents were informed, and their consent was obtained before the beginning of the questionnaire.

### 2.3. Data Analysis

SPSS 25.0 was used to perform the reliability test of the questionnaire [28], and AMOS 24.0 was used to build and validate the structural equation model [29]. According to hypotheses, H1–H5, 7 latent variables are set in this paper’s model (Table 1), and the research hypotheses are tested by analyzing the causal relationships between latent variables. The selection of potential variables and the hypothesis of the relationship between potential variables were mainly referred to the Health Belief Model [30] and related studies. The seven potential variables are digital media exposure, perceived susceptibility, perceived severity, perceived benefits, perceived barriers, cues to action, and intentions of influenza vaccination. Each latent variable was measured by 2–6 observed variables, expressed as specific question items in the questionnaire. The model was evaluated, and hypotheses were tested by analyzing data such as factor loadings (coefficients between observed and latent variables), path coefficients (coefficients between observed variables), and goodness of fit (coefficients to evaluate the overall fit of the model) of the model. 

## 3. Results

After the model was constructed, the valid samples were imported into AMOS 24.0 for model verification. Figure 1 shows the calculation results of the model. According to the model diagram and the software output results, the results were analyzed from three aspects: (1) Factor Load Analysis: Analyze the mean value and standard deviation of each observable variable and the load coefficient of the observable variable to the latent variable. (2) The Measurement Model: Calculate the correlation coefficient and evaluate each fitting index of the structural model. (3) The Structural Model: Interpret the model path coefficient and verify the research hypotheses.

### 3.1. Factor Load Analysis & The Measurement Model

Confirmatory factor analysis (CFA) assessed the reliability and validity of the constructs. As shown in Table 2, Cronbach’s alpha has ranged from 0.71 to 0.89, which was more significant than the threshold value of 0.70. Thus, all constructs have acceptable reliability. Furthermore, the convergent validity was tested by examining the value of factor loadings, composite reliability, and average variance extracted (AVE). Table 2 shows that all factor loadings reached the benchmark value, better than 0.7, and a small amount greater than 0.5 is acceptable [33]. Moreover, all the average variance extracted is more significant than the benchmark value of 0.5. All the composite reliability is more significant than the 0.7 benchmark value [33]. The results indicate that all constructs have good convergent validity [34].

Besides, the model fit indicators (x^2^ = 1010.14, df = 382, x^2^/df = 2.64, RMR = 0.04, GIF = 0.93, AGFI = 0.91; NFI = 0.903, IFI = 0.931, CFI = 0.940; RMSEA = 0.06) also reflect a good fit between the measurement model and the dataset. Therefore, the reliability and validity of this study were supported.

### 3.2. The Structural Model

Figure 1 presents the results of the hypotheses testing. The estimated parameters include path coefficients (β) and critical ratios (t values). Figure 1 shows that digital media exposure positively affects perceived susceptibility and vaccination intentions (H1: β = 0.78, t = 5.21; β = 0.13, t = 4.02), thus supporting H1. Digital media exposure positively affects perceived severity with a high path coefficient, and perceived severity also positively affects vaccination intentions (H2: β = 0.88, t = 7.87; β = 0.20, t = 5.11). Thus, H2 is supported. Digital media exposure positively affects perceived benefits, and perceived benefits positively affect vaccination intentions (H3: β = 0.59, t = 3.32; β = 0.41, t = 10.52), thus supporting H3. In addition, digital media exposure negatively affects perceived barriers, and perceived barriers negatively affect vaccination intentions (H4: β = −0.26, t = −4.03; β = −0.27, t = −5.37), thus supporting the H4. Digital media exposure positively affects cues to action, and cues to action positively affect vaccination intentions (H5: β = 0.74, t = 5.17; β = 0.60, t = 4.78), which indicates that the H5 is also supported. All the t values are more significant than 3.29, which suggests that the significance of the coefficients of ten paths all reached the benchmark value of 0.001. So, all the five hypotheses in this study are supported.

## 4. Discussion

This study examined the association between information exposure to digital media and influenza vaccination intentions. Based on the HBM model and the three-stage model, the theoretical model was constructed by introducing the mediating variables of perceived susceptibility, perceived severity, perceived benefits, perceived barriers, and cues to action to represent health beliefs [24]. This paper systematically evaluates how digital media exposure affects the vaccination intention of influenza vaccines through these mediating variables. The results show that all the five hypotheses proposed in the theoretical model are supported. The theoretical model explains the influence of the Chinese people’s digital media exposure on vaccination against influenza.

Among them, digital media exposure has the most significant influence on perceived susceptibility and perceived severity. However, the path coefficient of these two mediating variables and vaccination intentions is not high. The reason may be that although some people are aware of the seriousness of influenza and the possible threat of not being vaccinated through information contact, they lack specific cues to action [25].

Consistent with other research results, digital media exposure has a more significant impact on perceived benefits, which in turn has a more significant effect on vaccination intentions [1]. Users who feel the benefits of influenza vaccination from digital media exposure are more inclined to be vaccinated. Digital media exposure can also reduce the public’s perceived barriers and improve vaccination intentions by popularizing the necessity of vaccination.

It suggests that when using digital media platforms to promote vaccination, on the one hand, it is necessary to encourage the public’s cognition of perceived benefits, significantly to enhance the public’s recognition of the effectiveness of influenza vaccines. On the other hand, it is necessary to release authoritative information about vaccine effectiveness, who needs to be vaccinated, and the necessity of vaccination to reduce the perceived barriers of the public and promote vaccination [32].

The research also shows that the path coefficient of ‘information-cues to action-vaccination intentions’ is very high, indicating the importance of mediator cues to action in promoting vaccination. Digital media information can be an essential in promoting action, thus influencing the public’s intentions to vaccinate. Previous studies have also confirmed that cues to action can directly affect users’ intentions to influenza vaccination [26]. Therefore, digital media platforms should be fully utilized in vaccine promotion to carry out cues to vaccination action. The content advocating public vaccination of influenza vaccine should be disseminated suitable for digital media platforms.

Vaccination is one of the most critical components of public health programs. It plays an essential role in curbing the prevalence of infectious diseases [35], and one of the main challenges facing public health systems is ensuring adequate vaccination coverage [36]. Therefore, this study also inspires public health agencies at the international and national levels to improve vaccination rates and promote public health. Given the critical role of digital media in vaccination, the active role of digital media cannot be ignored at the level of communication strategies when promoting different vaccination campaigns, including influenza vaccination [7,10]. Particularly affected by the COVID-19 pandemic, traditional interpersonal communication influence is weakened, so it is more important to pay attention to the vital role of digital media in public crisis [19]. However, it should be noted that differences in the credibility of different media can affect the public’s perceived risk [37]. Therefore, relevant public health agencies should focus on their credibility when using digital media to promote vaccination campaigns.

Previous studies have explored traditional media on vaccination intentions, such as newspapers and television [15,16]. With the penetration of digital media into our lives, we have extended our research scope to digital media, which is more in line with the need, for vaccine promotion in the current information society. Previous studies have focused on website information content on vaccination intentions [38]. This study explored the impact on vaccination intentions from users’ perspectives in a broader scope of digital media exposure.

However, study limitations must be noted. Firstly, the questionnaire samples collected in this study generally have a high level of education. Whether the conclusions drawn in this study are also applicable to people with a low level of education remains to be further discussed. Secondly, the study examined perceived susceptibility, perceived severity, perceived benefits, perceived barriers, and cues to action as mediating factors. However, there may be other mediating factors between digital media exposure and influenza vaccination intentions, such as healthy self-efficacy, etc. [21]. These factors are expected to be supplemented in subsequent studies.

## 5. Conclusions

Given China’s large population base, it is unrealistic to implement a full-scale free influenza vaccination in China [39]. With the increasing influence of digital media on people’s lives, it is necessary to explore how digital media affects the public’s willingness to vaccinate. This study found that exposure to information about the influenza vaccine in digital media can influence people’s health beliefs and their intention to receive influenza vaccination. It also validates the potential function of digital media in vaccine promotion, etc. At the same time, this study can inspire public health institutions to innovate vaccine science strategies and contribute experiences to the construction of public health safety in various countries.

## Figures and Tables

**Figure 1 vaccines-10-01913-f001:**
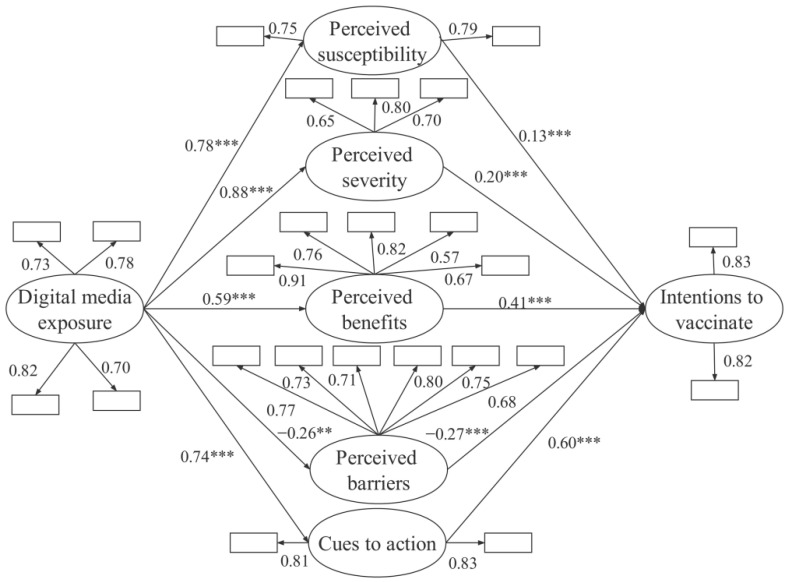
Results from the structural equation modeling procedure for the final model: Values indicate standardized regression weights and the path load factor. Notes: ** *p* < 0.01, t > 2.58; *** *p* < 0.001, t > 3.29.

**Table 1 vaccines-10-01913-t001:** Constructs and items included in the questionnaire.

Construct	Item	Reference
Digital Media Exposure	I have been exposed to information about the influenza vaccine on News websites/apps (Toutiao, Tengxun News, People’s Daily, etc.).	[21]
	I have contact information about the influenza vaccine on SNS websites/apps (WeChat, Weibo, Xiaohongshu, etc.).	[21]
	I have contact information about the influenza vaccine on online community websites/apps (Douban, Zhihu, Tianya, etc.).	[21]
	I have been exposed to flu vaccines on online video websites/short video apps (Bilibili, Douyin, Kuaishou, etc.).	[21]
Perceived susceptibility	I am worried about catching the flu in the fall and winter.	[1]
	I think I am more likely to get the flu than other people.	[26]
Perceived severity	The flu is a severe illness to me.	[1]
	If I get the flu, it will seriously affect my daily life, work, or study.	[26]
	If I get the flu accidentally, it will be a health threat to the whole family.	[1]
Perceived benefits	Vaccination is the most effective way to prevent influenza.	[26]
	Vaccination can alleviate the symptoms after infection, even if it can’t wholly avoid infection with the influenza virus.	[6]
	Vaccination can avoid the possible loss of work, time, energy, and economy caused by influenza.	[31]
	Vaccination against influenza can avoid the risk of my family catching influenza because of me.	[1]
	Vaccination can reduce my fear of catching influenza and play a significant role in psychological comfort.	[1]
Perceived barriers	The flu virus keeps mutating, and I doubt the effectiveness of the influenza vaccine.	[1]
	Healthy people can also prevent influenza with proper daily protection, and vaccination is not necessary.	[32]
	Getting an influenza vaccine is not conducive to the establishment of immunity and resistance to influenza.	[26]
	Influenza is not a severe and life-threatening illness, and patients usually recover within one to two weeks.	[26]
	I don’t know “I need to be vaccinated against influenza.”	[31]
	I don’t know how to apply for influenza vaccination.	[31]
Cues to action	I have received messages from social media urging everyone to get an influenza vaccine.	[1]
	The bad news about the flu epidemic on social media also influenced my decision to influenza vaccination.	[15]
Vaccination intentions	If conditions permit, I am willing to vaccinate against influenza.	[31]
	If conditions permit, I will make a plan for influenza vaccination in the future.	[31]

**Table 2 vaccines-10-01913-t002:** Factor Load Analysis & Reliability and Validity Test.

Construct	Item	Loading	M	SD	Cronbach’s Alpha	Composite Reliability	AVE
Digital media exposure	DMP1	0.73	3.29	0.81	0.71	0.84	0.58
	DMP2	0.78					
	DMP3	0.82					
	DMP4	0.70					
Perceived susceptibility	PS1	0.75	3.33	0.91	0.87	0.74	0.59
	PS2	0.79					
Perceived severity	PSR1	0.65	3.38	0.90	0.77	0.76	0.52
	PSR2	0.80					
	PSR3	0.70					
Perceived benefits	PB1	0.91	3.59	0.91	0.89	0.87	0.57
	PB2	0.76					
	PB3	0.82					
	PB4	0.57					
	PB5	0.67					
Perceived barriers	PBR1	0.77	3.32	0.89	0.73	0.88	0.55
	PBR2	0.73					
	PBR3	0.71					
	PBR4	0.80					
	PBR5	0.75					
	PBR6	0.68					
Cues to action	CTA1	0.81	3.42	0.98	0.71	0.80	0.67
	CTA2	0.83					
Intentions to vaccinate	ATV1	0.83	3.31	0.71	0.79	0.81	0.68
	ATV2	0.82					

## Data Availability

The raw data supporting the conclusions of this article will be made available by the authors, without undue reservation.

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
