# Peer review of "Digital Media Exposure and Health Beliefs Influencing Influenza Vaccination Intentions: An Empirical Research in China"

_vaccines, 2022, doi:10.3390/vaccines10111913_

Round 1
Reviewer 1 Report
The manuscript written by Yin and colleagues focuses on the effect of the digital media exposure on the influenza vaccination intentions in China. While the topic is of interest, there are some major flaws which in my opinion do not allow me to accept the manuscript for publication in the Vaccines. Please see below my detailed comments:
1. Yin and colleagues collected only 600 valid questionnaires. It is not enough to conclude anything for the Chinese population. Did authors calculate the minimum number of participants to fulfill the representative group criteria for China? Which statistical test were used to analyze the data?
2. In the questionnaire authors should ask on the influenza vaccination in the previous years? This would be very helpful information which potentially may confirm the influenza vaccination intentions.
3. What do the authors mean about the potential bias related to the algorithm and cookies used by the digital media platforms which showed the users similar topic to the previously searched ones?
4. Finally, I would like to ask how did authors deal with the limited accessibility to the digital media in the senior groups? This aspect needs to be discussed.
Taking into account the above comments, with regret I need to recommend rejection of the manuscript in the current form.
Author Response
Dear reviewer:
We appreciate your careful review of this manuscript. You have made many constructive comments and made us aware of the shortcomings of the article and the path to improving it. Based on your suggestions, we have fully reflected on the manuscript and thoroughly revised it. Following the journal's return requirements, we have listed the reviewers' comments and our responses in the table below. We hope our modifications will satisfy you.
- Yin and colleagues collected only 600 valid questionnaires. It is not enough to conclude anything for the Chinese population. Did the authors calculate the minimum number of participants to fulfill the representative group criteria for China? Which statistical tests were used to analyze the data?
Thank you to the reviewers for reviewing this article. We have revised the article according to your suggestions and would like to respond to some of your queries here.
1. More than 800 samples were initially collected in this paper, and 600 valid samples were retained after reliability testing. From the existing studies on "Chinese people's perceptions of vaccines," the number of questionnaires reached 500, which indicates that the study sample is representative(Kundan et al., 2022; Wan et al., 2021). Therefore, our 600 valid samples can reflect a certain extent, the knowledge of Chinese people about the influenza vaccine and its vaccination efforts, which is representative. We have modified the sample collection process in the methods section to highlight the representation of the samples in response to your question.
Kundan, Z., Hossain, M. S., & Khalifa, G. S. A. (2022). Factors Determining the COVID-19 Vaccinated Tourists' Intention to Repeat Behaviour: An Empirical Perspective for a New Normal. Sustainability, 14(21), 13888.
Wan, X., Huang, H., Shang, J., Xie, Z., Jia, R., Lu, G., & Chen, C. (2021). Willingness and influential factors of parents of 3-6-year-old children to vaccinate their children with the COVID-19 vaccine in China. Human Vaccines & Immunotherapeutics, 17(11), 3969-3974.
2.We use structural equation modeling to verify the relationship between the different variables. We set 7 latent variables, each measured by 2-6 observed variables. We created various types of variables and built variable relationships in AMOS software, and then imported the data to automatically validate the structural equation model. The model automatically generates the causal path coefficients between the potential variables. We further explain the variable settings and model tests in the methods section, which clarifies the statistical testing process.
- In the questionnaire, the authors should ask about influenza vaccination in previous years. This would be very helpful information which potentially may confirm the influenza vaccination intentions.
We thank the reviewers for pointing out the problems and suggestions regarding the question set in the questionnaire. We asked about prior influenza vaccination status and knowledge about influenza vaccine in our questionnaire. For example, Q7: Have you ever had the flu? Q8: Please select the correct option for the following "general knowledge of flu illness" [Multiple choice] Q9: Which of the following descriptions of influenza illnesses matches your opinion? Q10: Have you ever received a flu vaccination? Q11: Please select the option you are aware of for the following information about the flu vaccine [multiple choice]. Q12: Which of the following descriptions of the flu vaccine matches your opinion?
From the recall results, most of the respondents had not received the influenza vaccine, so there is still room for promoting and popularizing the influenza vaccine in China. It also provides a premise for our study to be conducted.
- What do the authors mean about the potential bias related to the algorithm and cookies used by the digital media platforms which showed the users similar topic to the previously searched ones?
On the question of whether technologies such as algorithms for digital platforms affect access to information on digital platforms. First, in China nowadays, people have access to information on various digital platforms, and we cannot break down the specific impact of a particular digital platform on human cognition or behavior. Second, on the issue of access to health information about influenza vaccines, although algorithmic mechanisms can influence people's access to information, we should still emphasize the initiative of people themselves because there is no way to directly measure the impact of algorithms on people's access to information. Finally, according to the specific context of China, we have divided the media into traditional and new media, which is more in line with Chinese media usage habits.
- Finally, I would like to ask how authors deal with the limited accessibility to digital media in the senior groups. This aspect needs to be discussed.
On the exposure of digital media for the elderly. With the increasing popularity of digital media in China, the impact of digital media has penetrated different age groups, including the elderly. Some studies show that more older adults use digital media (Zhou et al., 2022). In this context, we do not specifically distinguish between the digital media use of older and other age groups. In addition, in our valid sample, the elderly over 60 occupy 7% of the total sample, a proportion that has also been quantified according to the 2010 census data of the National Bureau of Statistics of China.
Zhou, Y., He, T., & Lin, F. (2022). The Digital Divide Is Aging: An Intergenerational Investigation of Social Media Engagement in China. International Journal of Environmental Research and Public Health, 19(19), 12965.
Reviewer 2 Report
This is an interesting manuscript by Yin and colleagues which aims to investigate whether/how digital media exposure influences people's intention to influenza vaccination.
It's an important topic as influenza vaccination coverage rates around the world are low and influenza susceptibility is likely increasing in the general population after 2 years of low influenza activity; therefore, tools to understand how to reduce vaccine hesitancy and increase coverage rates are welcome.
Some comments:
- The introduction is nicely written. I only suggest adding a short context paragraph to introduce the topic of the influence exerted by digital media not only for influenza, but also for other VPD (as COVID-19). I think this would be important to explain that this issues are often common when it comes to VPDs. Here some references you might want to use [1-3].
- I suggest better explaining the 5 hypoteses presented, in particular by making explicit the second verb in the sentence. Otherwise I think it can be not very clear to a reader.
E.g. H3 "Digital media exposure positively affects perceived benefits and POSITIVELY AFFECTS intentions to vaccinate".
H4 "Digital media exposure negatively affects perceived barriers and NEGATIVELY AFFECTS intentions to vaccinate"
- In the methods section, I suggest better explaining how you build the model and why you use those specific hypoteses. Have you tested the model with different hypoteses?
- In the discussion section, I suggest adding a paragraph where you better explain the implications for public health that your model will have. How could this change public health practices?
1. Del Riccio M, Bechini A, Buscemi P, Bonanni P, On Behalf Of The Working Group Dhs, Boccalini S. Reasons for the Intention to Refuse COVID-19 Vaccination and Their Association with Preferred Sources of Information in a Nationwide, Population-Based Sample in Italy, before COVID-19 Vaccines Roll Out. Vaccines (Basel). 2022 Jun 8;10(6):913. doi: 10.3390/vaccines10060913. PMID: 35746521; PMCID: PMC9229641.
2. Lee J, Choi J, Britt RK. Social Media as Risk-Attenuation and Misinformation-Amplification Station: How Social Media Interaction Affects Misperceptions about COVID-19. Health Commun. 2021 Nov 9:1-11. doi: 10.1080/10410236.2021.1996920. Epub ahead of print. PMID: 34753361.
3. Teoh D. The Power of Social Media for HPV Vaccination-Not Fake News! Am Soc Clin Oncol Educ Book. 2019 Jan;39:75-78. doi: 10.1200/EDBK_239363. Epub 2019 May 17. PMID: 31099637.
Author Response
Dear reviewer:
We appreciate your careful review of this manuscript. You have made many constructive comments and made us aware of the shortcomings of the article and the path to improving it. Based on your suggestions, we have fully reflected on the manuscript and thoroughly revised it. Following the journal's return requirements, we have listed the reviewers' comments and our responses in the table below. We hope our modifications will satisfy you.
- The introduction is nicely written. I only suggest adding a short context paragraph to introduce the topic of the influence exerted by digital media not only for influenza but also for other VPD (as COVID-19). I think this would be important to explain that this issues are often common when it comes to VPDs. Here some references you might want to use [1-3].
Thanks for recommending the literature for our study, which inspired us when revising this paper, and we referred to it in the revised paper.Concerning the literature provided by the reviewers, we added the impact of information exposure to digital media on the covid-19 vaccine and HPV vaccination (mainly in the introductory section) and integrated the relevant paragraphs.
- I suggest better explaining the 5 hypotheses presented, in particular by making explicit the second verb in the sentence. Otherwise, I think it cannot be very clear to a reader.
We have taken your comments seriously and revised the relevant statements in the assumptions section to make our presentation more understandable to readers.
- In the methods section, I suggest better explaining how you build the model and why you use those specific hypotheses. Have you tested the model with different hypotheses?
The variables, hypotheses, and models in this paper are mainly referred to as the Health Belief Model and some existing studies. As suggested, we further explain the model (hypothesis) construction process and add some details in the methods section.
We first proposed variables and hypotheses based on existing studies and Health Belief Models. And then, the model was constructed based on the variables and hypotheses. We also set the questionnaire items according to the model and conducted the research work. Finally, the model was validated based on the returned questionnaires. Since the model itself was set up based on the hypothesis, the model's validation also proved the hypothesis's validation.
- In the discussion section, I suggest adding a paragraph where you better explain the implications for public health that your model will have. How could this change public health practices?
We thank the reviewers for your suggestion to state the impact of this paper on public health in the discussion section. We have carefully drawn your comments and read the relevant literature. Based on this, we added the implications of this paper for public health at the international and national levels, such as the role digital media can play in public health campaigns to enhance vaccination in the discussion section.
Round 2
Reviewer 1 Report
I'm satisfied with the author's comment and recommend the m,anuscript for publication.
Reviewer 2 Report
Many thanks for addressing the issues I had reported.
I have no further comments.